# OBJECT-CENTRIC NEURAL SCENE RENDERING

## ABSTRACT

We present a method for composing photorealistic scenes from captured images of objects. Our work builds upon neural radiance fields (NeRFs), which implicitly model the volumetric density and directionally-emitted radiance of a scene from a collection of images. While NeRFs synthesize realistic pictures, they only model static scenes and are closely tied to specific imaging conditions. This property makes NeRFs hard to generalize to new scenarios, including new lighting or new arrangements of objects. Instead of learning a scene radiance field as a NeRF does, we propose to learn object-centric neural scattering functions (OSFs), a representation that models per-object light transport implicitly using a lighting- and view-dependent neural network. This enables rendering scenes even when objects or lights move, without retraining. Combined with a volumetric path tracing procedure, our framework is capable of rendering light transport effects including occlusions, specularities, shadows, and indirect illumination, both *within* individual objects and *between* different objects. We evaluate OSFs on synthetic and real world datasets, and on generalizing to new scene configurations. Learning OSFs leads to photorealistic, physically-accurate renderings of multi-object scenes.

## 1 INTRODUCTION

Synthesizing images of dynamic scenes is an important problem in computer vision and graphics, with applications in AR/VR and robotics (Savva et al., 2019; Xia et al., 2020). For synthetic scenes, a user typically designs a set of 3D objects separately, then composes them into scenes to be rendered with specified camera, material, and lighting parameters. While this traditional graphics approach allows for flexible scene compositions, it requires detailed models of geometry, lighting, materials, and cameras, which can be difficult to obtain for real-world scenes.

To render real-world scenes without computer graphics models, recent works have explored using neural implicit methods (Lombardi et al., 2019; Sitzmann et al., 2019a;b). Most notably, Mildenhall et al. (2020) proposed neural radiance fields (NeRF), which achieve photorealistic quality by implicitly modeling the volumetric density and directional emitted radiance of a scene.

However, as shown in Figure 1, NeRF cannot generalize beyond the scene it was trained on, because it assumes static scenes and fixed illumination and learns a radiance field, which estimates only the resulting radiance along a ray after all light transport has occurred in a scene. Thus, for dynamic scenes where lights and objects can move, a separate NeRF-based model is needed for each new scene configuration.

(a) NeRF (Baseline)  (b) OSF (Our Method)

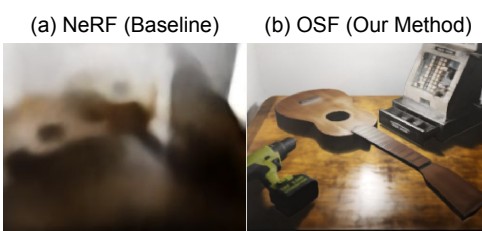

Figure 1: (a) NeRF. (b) Our method.

To address this issue, we propose Object-Centric Neural Scattering Functions (OSFs) to synthesize dynamic scenes of objects learned from 2D images (Figure 2). We represent each object as a learned 7D scattering function with inputs $(x, y, z, \phi_i, \theta_i, \phi_o, \theta_o)$, where $(x, y, z)$ is the spatial location, $(\phi_i, \theta_i)$ is the incoming light direction, and $(\phi_o, \theta_o)$ is the outgoing light direction. The function outputs the volumetric density as well as the fraction of light arriving from direction $(\phi_i, \theta_i)$ that scatters in outgoing direction $(\phi_o, \theta_o)$.

Each OSF models all light bounces (reflections) and occlusions (shadows) within an object. Since each object's scattering function is a radiance transfer function rather than a radiance field, it is

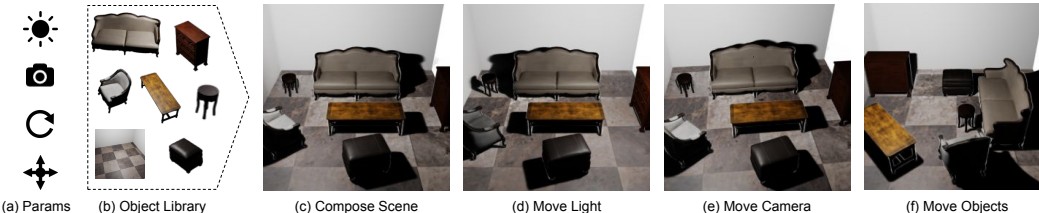

(a) Params    (b) Object Library    (c) Compose Scene    (d) Move Light    (e) Move Camera    (f) Move Objects

Figure 2: We propose an object-centric neural scene representation for image synthesis. Given a scene description (a), and a repository of neural object-centric scattering functions (OSF) trained independently from images and frozen for each object (b), we can compose the objects into scenes (c), and render photorealistic images as we move lights (d), cameras (e), and/or objects (f). Our framework is capable of rendering occlusions, specularities, shadows, and indirect illumination.

intrinsic to the object (independent of the scene it is in) and can be reused across different object placements and lighting conditions without retraining. We emphasize that because NeRFs are radiance fields, they cannot be composed, and cannot generalize beyond *one* scene. In contrast, we can render infinitely many scenes. We can build a library of OSFs trained independently for different objects to be composed into scenes with different object placements, camera, and lighting.

To model light transport *between* objects, we integrate our implicit object functions with volumetric path tracing. Like NeRF, we evaluate the radiance and volumetric density at 5D samples along every primary ray to the camera and composite them with an over operator. However, unlike NeRF, we estimate the radiance for each 5D sample by integrating our 7D OSF across the 2D sphere of incoming light directions. We estimate the integral with Monte Carlo path tracing (Kajiya, 1986) to reproduce shadows and indirect illumination effects.

Our key idea is to decompose the rendering problem into (i) a learned component (per-object *asset creation*), and (ii) a non-learned component (per-scene *path tracing*). The learned component models intra-object light transport (e.g., bounces from the seat of a chair to the back of the chair). The non-learned component handles inter-object light transport (e.g., bounces from a wall to a chair). Together, they model the full rendering equation (Kajiya, 1986) (except for occluders or light sources that intrude the object's convex hull (Sloan et al., 2002)). Since only the inter-object light transport changes as objects and lights move, no re-training is required for different scene arrangements. Experimental results indicate that our method is capable of rendering images with novel scene compositions and lighting conditions better than alternative learned approaches.

In summary, our contributions are:

1. Learning Object-Centric Neural Scattering Functions (OSFs) that model intra-object light transport implicitly using a lighting- and view-dependent neural network.
2. Integrating implicitly learned object scattering functions with volumetric path tracing to model inter-object light transport.
3. A rendering algorithm that enables rendering scenes with moving objects, lights and cameras, using implicit functions.

## 2 RELATED WORK

**Classical object-centric representations.** Factoring light transport into intra- and inter-object illumination has a long history in traditional computer graphics (Dutre et al., 2018). In most cases, the motivation is to improve rendering efficiency by approximating intra-object lighting factors with simple transfer functions (e.g., linear) for simple radiance fields (e.g., spherical harmonics) derived from from computer graphics models, as in precomputed radiance transfer (PRT) (Sloan et al., 2002), ambient occlusion (Miller, 1994), or virtual walls (Arnaldi et al., 1994). In other cases, the motivation is to insert captured, real-world radiance fields into synthetic scenes, as in Light Field Transfer (Cossairt et al., 2008). These methods generally store the radiance field for objects in a discrete representation (e.g., a sampled 2D or 4D grid). As a result, they cannot reproduce accurate inter-object light transport, especially for objects with intersecting bounding volumes. In contrast, we focus on learning radiance transfer from images in order to model complex real-world scattering accurately, and utilize volumetric rendering techniques to account for inter-object illumination.

**Novel view synthesis.** Traditional methods for synthesizing novel views of a scene from captured images include using Structure-From-Motion (Hartley & Zisserman, 2003) and bundle adjustment (Triggs et al., 1999) to predict a sparse point cloud and camera parameters of the scene. More recently, a number of learning-based novel view synthesis methods have been presented but require 3D geometry as inputs (Hedman et al., 2018; Thies et al., 2019; Meshry et al., 2019; Aliev et al., 2020; Martin-Brualla et al., 2018). Others use multiplane images as proxies for novel view synthesis, but their viewing ranges are limited to interpolated input views (Flynn et al., 2016; Zhou et al., 2018; Srinivasan et al., 2019; Mildenhall et al., 2019). Some works represent scenes as coarse voxel grids and use a CNN-based decoder for differentiable rendering, but lack view consistency due to the use of 2D convolutional kernels (Nguyen-Phuoc et al., 2018; 2019; 2020).

Recently, volume rendering approaches have been used to render scenes represented as voxel grids that are more view-consistent (Lombardi et al., 2019; Sitzmann et al., 2019a). However, the rendering resolution of these methods are limited by the time and computational complexity of discretely sampled volumes. To address this issue, Neural Radiance Fields (NeRF) (Mildenhall et al., 2020) directly optimizes a *continuous* radiance field representation using a multi-layer perceptron. This allows synthesizing novel views of realistic images at an unprecedented level of fidelity. To make NeRF more efficient, Neural Sparse Voxel Fields (Liu et al., 2020) have been proposed as a sparse voxel octree variant of NeRF and demonstrate the ease of composing learned NeRFs with their voxel representation. See (Dellaert & Yen-Chen, 2020) for survey. While these implicit methods produce high-quality novel views of a scene, their models assume a static scene with fixed illumination. Our method enables synthesizing dynamic scenes with novel viewpoint, lighting, and object configurations.

**Relighting.** Learning-based methods that relight images without explicit geometric reasoning have been proposed, but lack the ability to recover hard shadows (Sun et al., 2019; Xu et al., 2018; Zhou et al., 2019). Other works use geometric representations that facilitate shadowing computation, but require 3D geometry as input (Philip et al., 2019; Zhang et al., 2021; Oechsle et al., 2020; Rematas & Ferrari, 2020). Deep Reflectance Volumes (Bi et al., 2020b) reconstructs a voxelized representation of a scene and predict per-voxel BRDFs, but the fixed resolution of voxel grids limits the quality in the rendered images. Similarly, Neural Reflectance Fields (Bi et al., 2020a) predicts the parameters of a BRDF model, but demonstrate higher fidelity rendering by learning a continuous scene representation. However, Neural Reflectance Fields focuses on relighting single objects, and requires manual specification of the BRDF model. Parametric BRDF models are unable to handle complex scattering functions, including real-world scattering phenomena that are difficult to model. In contrast, our method is capable of learning *all* scattering functions, and can render *multiple* objects in dynamic scenes.

## 3 PRELIMINARIES

### 3.1 VOLUME RENDERING

To render an image of a scene with arbitrary camera parameters, camera rays are sent into the scene, through each pixel on the image plane. The expected color of each pixel is computed as the radiance along each camera ray.

Volume rendering is an approach for computing the radiance traveling along rays traced in a volume. Let $r(t) = x_0 + \omega_o t$ be a point along a ray $r$ with origin $x_0$ and direction $\omega_o$, where $t \in \mathbb{R}$ is a 1D location along the ray, and the $o$ in $\omega_o$ denotes "outgoing" direction. For our purposes, we assume non-emissive and non-absorptive volumes. From Novák et al. (2018), the volume rendering equation to compute the radiance $L(x_0, \omega_o)$ of the ray is defined as:

$$L(x_0, \omega_o) = \int_{t_n}^{t_f} \tau(t)\sigma(r(t))L_s(r(t), \omega_o)\,dt, \quad \text{where} \quad \tau(t) = \exp\left(-\int_{t_n}^{t} \sigma(r(u))\,du\right), \quad (1)$$

where $t_n$ and $t_f$ are near and far integration bounds, $\sigma(r(t))$ denotes the volume density of point $r(t)$, and $\tau(t)$ denotes the accumulated transmittance from $t_n$ to $t$. The term $L_s(r(t), \omega_o)$ is the light scattered at point $r(t)$ along direction $\omega_o$, defined as the integral over all incoming light directions:

$$L_s(x, \omega_o) = \int_{\mathcal{S}} L(x, \omega_l) f_p(x, \omega_l, \omega_o)\,d\omega_l, \quad (2)$$

where $\mathcal{S}$ is a unit sphere and $f_p$ is a phase function that evaluates the fraction of light incoming from direction $\omega_l$ at a point $x$ that scatters out in direction $\omega_o$. In NeRF, Mildenhall et al. (2020) assume

fixed illumination and do not consider any form of Equation 2. We consider a more general form of the volume rendering equation that explicitly models light paths within and between objects. This is important for dynamic scenes, where lighting and objects can move with respect to one another.

## 3.2 RAY MARCHING

The continuous integrals in Equation 1 can be estimated with quadrature (Kniss et al., 2003; Max, 1995), as done in NeRF (Mildenhall et al., 2020). For each ray, stratified sampling is used to obtain $N$ samples $\{t_i\}_{i=1}^N$ along the ray, where $t_i \in [t_n, t_f]$. The rendering equation is approximated by:

$$L(\boldsymbol{x}_0, \boldsymbol{\omega_o}) = \sum_{i=1}^N \tau_i \alpha_i L_s(\boldsymbol{x}_i, \boldsymbol{\omega_o}) \quad \text{where} \quad L_s(\boldsymbol{x}_i, \boldsymbol{\omega_o}) = \frac{1}{|\mathcal{L}|} \sum_{l \in \mathcal{L}} L(\boldsymbol{x}_i, \boldsymbol{\omega_l}) \boldsymbol{\rho}_i^{\boldsymbol{l}}, \quad (3)$$

where $\tau_i = \prod_{j=1}^{i-1}(1 - \alpha_j)$ and $\alpha_i = 1 - e^{-\sigma_i(t_{i+1}-t_i)}$. To compute the average over incoming light paths $L_s$, we discretize over the domain $\mathcal{S}$ in Equation 2 by sampling a set of incoming light paths $\mathcal{L} = \{\boldsymbol{l}_1, \ldots, \boldsymbol{l}_K\}$, where $\boldsymbol{\rho}_i^{\boldsymbol{l}} = f_p(\boldsymbol{x}_i, \boldsymbol{\omega_l}, \boldsymbol{\omega_o}) \in [0, 1]$, the fraction of light incoming from light path $\boldsymbol{l}$ that is scattered in direction $\boldsymbol{\omega_o}$.

## 3.3 NEURAL RADIANCE FIELDS

NeRF represents a continuous scene as a volumetric radiance field, approximated with a multilayer perceptron $F_\Theta$. The model $F_\Theta$ takes spatial location $\boldsymbol{x} = (x, y, z)$ and viewing direction $\boldsymbol{d} = (\phi, \theta)$ as input, and outputs the density $\sigma$ and color $\boldsymbol{c} = (r, g, b)$, where $r, g, b \in [0, 1]$. Frequency-based positional encoding (Rahaman et al., 2019; Vaswani et al., 2017) is applied to the inputs to better capture high-frequency variation in appearance and geometry.

A hierarchical volume sampling procedure (Mildenhall et al., 2020; Levoy, 1990) is then employed to more efficiently allocate samples along each ray. This technique biases sample allocation to favor the visible parts of the scene that contribute the most to the final render, avoiding occluded or free space in the scene. NeRF simultaneously optimizes two radiance fields, where the sample weights $\tau_i \cdot \alpha_i$ from a *coarse* model are used to bias samples for a *fine* model. The $L_2$ loss is used to optimize both models: $\sum_{\boldsymbol{r} \in \mathcal{R}} \|\widehat{C}_c(\boldsymbol{r}) - C(\boldsymbol{r})\|_2^2 + \|\widehat{C}_f(\boldsymbol{r}) - C(\boldsymbol{r})\|_2^2$, where $\mathcal{R}$ is the set of all camera rays, $\widehat{C}_c(\boldsymbol{r})$ and $\widehat{C}_f(\boldsymbol{r})$ denote the radiance along ray $\boldsymbol{r}$ predicted by the coarse and fine models respectively, and $C(\boldsymbol{r})$ is the ground truth pixel color for $\boldsymbol{r}$.

## 4 METHOD

### 4.1 OBJECT-CENTRIC NEURAL SCATTERING FUNCTION

We represent each object as a 7D object-centric neural scattering function (OSF), depicted in Figure 3a. For each object, we learn an implicit function $F_\Theta : (\boldsymbol{x}, \boldsymbol{\omega_l}, \boldsymbol{\omega_o}) \to (\sigma, \boldsymbol{\rho})$ that receives a 3D point in the object coordinate frame, the incoming light direction, and the outgoing light direction, and predicts the volumetric density as well as fraction of incoming light that is scattered in the outgoing direction. $\Theta$ are learned weights that parameterize the neural network, $\boldsymbol{x} = (x, y, z)$ denotes the spatial location, $\boldsymbol{\omega_l} = (\phi_l, \theta_l)$ denotes the incoming light direction, $\boldsymbol{\omega_o} = (\phi_o, \theta_o)$ denotes the outgoing light direction, $\sigma$ denotes the volumetric density, and $\boldsymbol{\rho} = (\rho_r, \rho_g, \rho_b)$ denotes the fraction of light arriving at $\boldsymbol{x}$ from direction $\boldsymbol{\omega_l}$ that is scattered and leaving in direction $\boldsymbol{\omega_o}$. The final color of a point $\boldsymbol{x}$ is the integral of $\boldsymbol{\rho}$ multiplied by the incoming radiance over all incoming light directions in unit sphere $\mathcal{S}$ (Equation 2). Following NeRF, we similarly apply positional encoding to our inputs $(\boldsymbol{x}, \boldsymbol{\omega_l}, \boldsymbol{\omega_o})$ and employ a hierarchical sampling procedure to recover higher quality appearance and geometry of learned objects.

During training, we assume a single point light source with radiance of $(1, 1, 1)$. This simplifies $L_s$ from Equation 2 to $L_s(\boldsymbol{x}, \boldsymbol{\omega_o}) = L(\boldsymbol{x}, \boldsymbol{\omega_l}) f_p(\boldsymbol{x}, \boldsymbol{\omega_l}, \boldsymbol{\omega_o}) = f_p(\boldsymbol{x}, \boldsymbol{\omega_l}, \boldsymbol{\omega_o})$. To learn per-object NeRFs independent of object rotation and translation, the inputs to $F_\Theta$ must be in the object's canonical coordinate frame. Given a object transformation $T_i$ for object $\boldsymbol{o}_i$, we apply $T_i^{-1}$ to $(\boldsymbol{r}, \boldsymbol{\omega_l}, \boldsymbol{\omega_o})$ before feeding the inputs to the network.

### 4.2 RENDERING MULTIPLE OSFS

Once we have learned an OSF for each object, we aim at composing the learned objects into scenes. An overview of our procedure is visually depicted in Figure 3b.

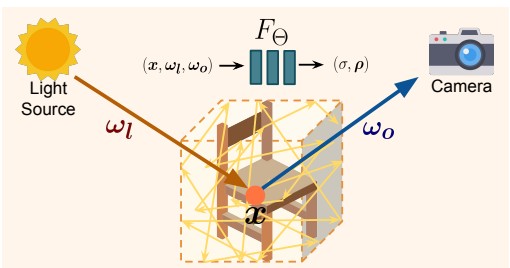 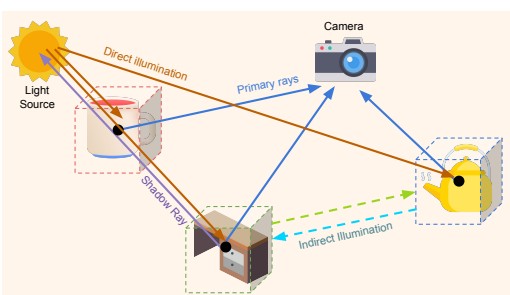

(a) We represent each object as an object-centric neural scattering function (OSF), which models how light entering at a point $\boldsymbol{x}$ on the object, from direction $\boldsymbol{\omega_l}$ where $\boldsymbol{l}$ corresponds to a light path, undergoes multiple bounces within the object and exits along direction $\boldsymbol{\omega_o}$ with some fractional amount of light $\boldsymbol{\rho}$. We approximate the scattering function with a multilayer perceptron $F_\Theta$ where $\Theta$ are learned weights that parameterize the neural network. Given a single point $\boldsymbol{x}$, an incoming light direction $\boldsymbol{\omega_l}$, and an outgoing direction $\boldsymbol{\omega_o}$, $F_\Theta$ outputs the volume density $\sigma$ of that point, as well as the fraction of light arriving at $\boldsymbol{x}$ from direction $\boldsymbol{\omega_l}$ that is scattered in direction $\boldsymbol{\omega_o}$.

(b) Our procedure for rendering an arbitrary scene consisting of multiple objects, light sources, and cameras. Given a set of objects, we compute direct illumination by shooting rays from each light source to each object (brown arrows). Shadows are computed by sending shadow rays back to each light source (purple arrow). The shadow ray from the desk is occluded by the mug, so the mug casts a shadow on the desk. We send secondary rays between objects to render indirect illumination effects, such as between the desk and the kettle (green and blue dashed arrows). Finally, rays are sent back to the camera to render the final image (dark blue arrows).

Figure 3: Using our method (OSFs) to render: (a) single and (b) multiple objects.

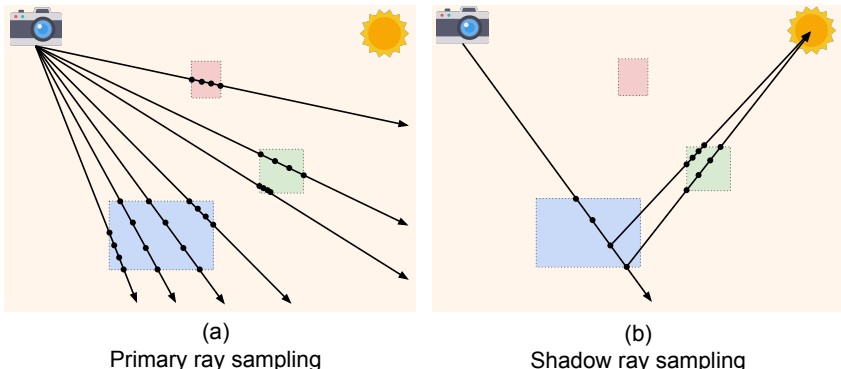

Figure 4: Sampling procedure. (a) Scene with a camera, light source, and object bounding boxes. Primary rays are sent from the camera into the scene. Rays that do not intersect with objects are pruned. Of the intersecting rays, we sample points within intersecting regions. (b) Shadow rays from each sample are sent to the light source, and samples within intersecting regions are evaluated.

Let $\mathcal{O} = \{\boldsymbol{o_i}\}_{i=1}^N$ be a set of $N$ objects we wish to render. For simplicity, we first describe the rendering process for each object $\boldsymbol{o_i}$, then explain the process to combine results across all objects to render the final scene. Let $\boldsymbol{o_i} \in \mathcal{O}$ denote object $i$ with transformation $T_i \in \mathbb{R}^{4 \times 4}$ and bounding box dimensions $D_i \in \mathbb{R}^3$. Further let $\boldsymbol{r}$ be a camera ray with origin $\boldsymbol{c} \in \mathbb{R}^3$ and direction $\boldsymbol{\omega_o} \in \mathbb{R}^3$, which we define with parameters $\gamma = [\boldsymbol{c}, \boldsymbol{\omega_o}] \in \mathbb{R}^6$. Our goal is to compute $L(\boldsymbol{c}, \boldsymbol{\omega_o})$ as described in Equation 3. We compute the ray-box intersection between the ray and the object to obtain near bound $t_n^i$ and far bound $t_f^i$ such that $\boldsymbol{r}(t_n^i)$ and $\boldsymbol{r}(t_f^i)$ each intersect a box plane, as shown in Figure 4. Note that rays that do not intersect with $\boldsymbol{o_i}$ are excluded from our computation. We sample $M$ points between $t_n^i$ and $t_f^i$ along ray $\boldsymbol{r}$ to obtain a sample $\boldsymbol{X}^i = \{\boldsymbol{x}_m^i\}_{m=1}^M$, where $\boldsymbol{X}^i \in \mathbb{R}^{M \times 3}$. Given a light source $\boldsymbol{l}$, we evaluate the object's model $F_{\Theta_i}(\boldsymbol{X}^i, \boldsymbol{\omega_l}, \boldsymbol{\omega_o})$ to obtain alpha values $\boldsymbol{\alpha}^i \in \mathbb{R}^M$ and phase function values $\boldsymbol{\rho}^i \in \mathbb{R}^{M \times 3}$.

It is not always possible for a light ray from light source $\boldsymbol{l}$ to reach the object $\boldsymbol{o_i}$. Any of the other objects in $\mathcal{O}' = \{\boldsymbol{o_j} \in \mathcal{O} \mid j \neq i\}$ in the scene may occlude the incoming light, casting a shadow on object $\boldsymbol{o_i}$. We compute shadows by sending a shadow ray $\boldsymbol{r}_m$ from each of the $M$ samples in $\boldsymbol{X}^i$

to the light source $l$. Evaluating the shadow ray enables us to determine the amount of light blocked along the ray by other objects. We define the parameters of the $M$ shadow rays as $\Gamma \in \mathbb{R}^{M \times 6}$.

For each object $\boldsymbol{o}_j \in \mathcal{O}'$, we compute ray-box intersections between shadow rays $\Gamma$ and $\boldsymbol{o}_j$'s bounding box. This allows us to compute the amount of light traveling towards $\boldsymbol{o}_i$ that is blocked by $\boldsymbol{o}_j$. Similar to primary rays, we sample $M$ points along each shadow ray to obtain a set of points $\boldsymbol{X}^j \in \mathbb{R}^{M \times M}$. We then evaluate the object model $F_{\Theta_j}(\boldsymbol{X}^j)$ to obtain alpha values $\boldsymbol{A}^j \in \mathbb{R}^{M \times M}$. For each shadow ray $\boldsymbol{r}_m$, we combine samples $\boldsymbol{A}_m^j$ across the $N-1$ objects in $\mathcal{O}'$ by sorting according to sample distance to obtain alpha values $\boldsymbol{A}_m \in \mathbb{R}^{M(N-1)}$. The fraction of unobstructed light traveling along the shadow ray $\boldsymbol{r}_m$ is computed as the transmittance:

$$\tau_m^l = \prod_{n=1}^{M(N-1)} (1 - \boldsymbol{A}_{mn}). \tag{4}$$

Thus, the adjusted incoming radiance from light source $l$ when accounting for occlusions is computed as $L_{\boldsymbol{l}}(\boldsymbol{x}_m, \boldsymbol{\omega_l}) = \tau_m^{\boldsymbol{l}} L_{\boldsymbol{l}}(\boldsymbol{x}_m, \boldsymbol{\omega_l})$.

We follow the scattering equation in Equation 2 and now consider all incoming light directions over the unit sphere $\mathcal{S}$. This accounts for secondary light rays traveling to an object $\boldsymbol{o}_i$ indirectly from another object $\boldsymbol{o}_j$ (indirect illumination). We approximate the integral over the unit sphere $\mathcal{S}$ by sampling $K$ directions on the unit sphere uniformly at random. For each direction $\boldsymbol{\omega_k}$ randomly sampled for a point $\boldsymbol{x}$, we send a secondary ray $\boldsymbol{r}_k$ from $\boldsymbol{x}$ in direction $\boldsymbol{\omega_k}$ and evaluate the radiance $L(\boldsymbol{x}, \boldsymbol{\omega_k})$ traveling along the ray. To compute the radiance of the secondary ray $L(\boldsymbol{x}, \boldsymbol{\omega_k})$, we employ the same technique used to compute the radiance of a primary ray $L(\boldsymbol{c}, \boldsymbol{\omega_o})$ (described at the beginning of Section 4.2). The incoming radiance $L(\boldsymbol{x}, \boldsymbol{\omega_k})$ is multiplied with the phase function value $\boldsymbol{\rho} = f_p(\boldsymbol{x}, \boldsymbol{\omega_k}, \boldsymbol{\omega_o})$ to determine the outgoing radiance $L(\boldsymbol{x}, \boldsymbol{\omega_o})$, where $\boldsymbol{\rho}$ is evaluated using $F_{\Theta_i}$. Note that this is possible due to the recursive nature of our formulation. Only secondary rays are described here (two bounces), but our method supports an arbitrary number of bounces.

**Rendering.** We sample and evaluate all objects in $\mathcal{O}$ to obtain alpha values $\{\boldsymbol{\alpha}^i\}_{i=1}^N$ and phase function values $\{\boldsymbol{\rho}^i\}_{i=1}^N$ for a set of sampled points $\{\boldsymbol{X}^i\}_{i=1}^N$ along ray $\boldsymbol{r}$. We sort the samples across all objects to produce a final set of $P = M \cdot N$ samples $\{\boldsymbol{x}_m\}_{m=1}^P$, $\{\alpha_m\}_{m=1}^P$, and $\{\boldsymbol{\rho}_m\}_{m=1}^P$.

Given light paths $\mathcal{L}$ containing both direct and indirect illumination, we render the final radiance of a ray with origin $\boldsymbol{x}_0$ and direction $\boldsymbol{\omega_o}$ with the following equation:

$$L(\boldsymbol{x}_0, \boldsymbol{\omega_o}) = \frac{1}{|\mathcal{L}|} \sum_{\boldsymbol{l} \in \mathcal{L}} \sum_{m=1}^P \alpha_m \boldsymbol{\rho}_m^{\boldsymbol{l}} \tau_m L_{\boldsymbol{l}}(\boldsymbol{x}_m, \boldsymbol{\omega_l}), \quad \text{where} \quad \tau_m = \prod_{n=1}^{m-1} (1 - \alpha_n), \tag{5}$$

and $L_{\boldsymbol{l}}(\boldsymbol{x}_m, \boldsymbol{\omega_l})$ is the radiance from light path $l$ arriving at point $\boldsymbol{x}_m$.

**Runtime.** In total, the cost of rendering a single image with $N_{\text{pixel}}$ pixels and $N_{\text{object}}$ objects is $\mathcal{O}(P^2 K N_{\text{pixel}})$. Note that $P$ is an upper bound on number of samples that need to be evaluated. In practice, a single ray often only intersects with at most one object in the scene, which means that the proposed rendering procedure is not significantly more expensive than the single object setting. We also note that compared to NRF Bi et al. (2020a) or traditional volumetric path tracing methods, OSF crucially does not require running path tracing *within* each object to simulate intra-object light bounces. This is because OSF learns the object-level scattering function that directly predicts the effects after all light bounces (reflections) and occlusions (shadows) within an object have occurred. Thus OSF is significantly faster than NRF which relies on simulating intra-object light bounces while querying its learned BRDF model.

In our experiments, rendering a single image with a single OSF at a resolution of $256 \times 256$ takes roughly 3.7 seconds. While the computation cost is high, there are efforts to reduce the rendering speed of NeRF that are orthogonal to this work. For instance, KiloNeRF (Reiser et al., 2021) can easily adapted to this work by utilizing thousands of tiny MLPs instead of one single large MLP to represent each OSF to obtain 1-2 orders of magnitude speed up.

## 5  EXPERIMENTS

**Datasets and evaluation metrics.** We evaluate our approach on several image datasets:

- FURNITURE-SINGLE: 15 objects rendered with random object pose, point light, and viewpoint.

- FURNITURE-RANDOM: 25 dynamic scenes, each containing a random layout of multiple objects, point light, and viewpoint.
- FURNITURE-REALISTIC: Scenes containing realistic arrangements of objects in rooms.
- REAL-NRF: Real-world objects from Bi et al. (2020a), captured in a dark room under varying viewing and lighting directions.
- REAL-OUTDOOR: Real-world outdoor scenes from Mildenhall et al. (2020).

For FURNITURE datasets, we use Blender's Cycles path tracer (Blender Foundation, 1994) to render images at $256 \times 256$ resolution for different object arrangements, camera views, and lighting configurations. We report PSNR, SSIM (Wang et al., 2003), and LPIPS (Zhang et al., 2018) metrics.

**Baselines and ablations.**    We compare our method to the following baselines:

1. **o-NeRF**: A variant of the NeRF model, but with one NeRF trained per object. When o-NeRFs are composed into scenes, they are rendered separately.
2. **o-NeRF + S**: An extension of o-NeRF with inter-object shadows; reduces the light arriving at each o-NeRF by the cumulative opacity of shadowing objects along the ray from the light (§4.2).

These baselines represent what could be achieved by combining separately trained NeRFs into a scene. Of course, since o-NeRFs produce radiance fields (not scattering fields), we do not expect them to perform well in novel lighting environments or object placements.

## 5.1  NOVEL LIGHTING

In the first experiment, we investigate how OSF method handles novel lighting conditions.

We train one model per object in FURNITURE-SINGLE. For each object model, we train on 400 images with randomized viewpoint and lighting, and test on 20 images of novel viewpoint and lighting. As can be seen in Figure 5, our method produces more accurate appearance of the objects in comparison to o-NeRF when tested on novel illumination conditions. In particular, o-NeRF fails to predict self-shadows for the couch and chair correctly. Additionally, o-NeRF fails to disentangle viewpoint versus lighting-dependent appearance, producing incorrect shadows for the couch and chair, and fails to capture the specular details of the ottoman. Quantitative results can be found in Table 1.

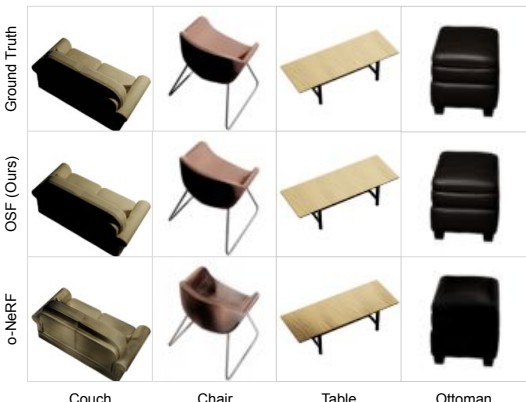

Figure 5: Novel lighting results.

## 5.2  SCENE COMPOSITION

In a second experiment, we conduct a scene composition task on FURNITURE-RANDOM, where multiple object models are combined into scenes in random pose, lighting, and viewpoint configurations. For this task, we use the same object models trained in Section 5.1. Results are shown in Table 1 and Figure 6. While not shown in the main text, results for FURNITURE-REALISTIC can be found in Appendix B.

Table 1: Quantitative results for novel lighting (FURNITURE-SINGLE) and scene composition (FURNITURE-RANDOM). Rows denote different methods: our full model (OSF), a variant of NeRF where one NeRF is trained per object (o-NeRF), and o-NeRF with shadows (o-NeRF + S).

| Dataset | FURNITURE-SINGLE | | | FURNITURE-RANDOM | | |
|---|---|---|---|---|---|---|
| Method | PSNR↑ | SSIM↑ | LPIPS↓ | PSNR↑ | SSIM↑ | LPIPS↓ |
| o-NeRF | 33.22 | 0.980 | 0.021 | 12.17 | 0.690 | 0.280 |
| o-NERF + S | — | — | — | 14.70 | 0.697 | 0.267 |
| OSF (Our Method) | **44.07** | **0.998** | **0.002** | **19.02** | **0.793** | **0.135** |

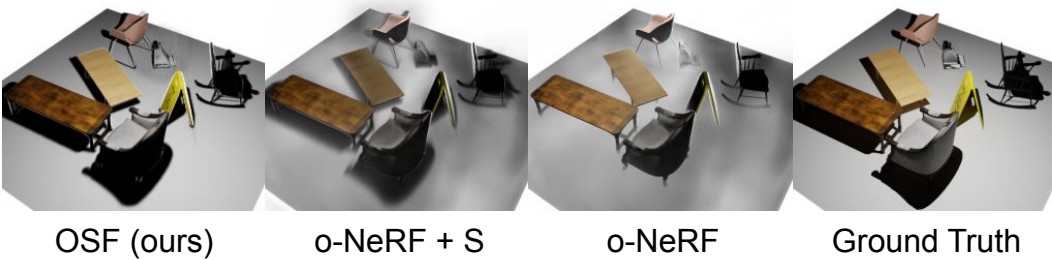

| OSF (ours) | o-NeRF + S | o-NeRF | Ground Truth |

Figure 6: Scene composition results on FURNITURE-RANDOM. The models OSF, o-NeRF, and o-NeRF + S are explained in §5. Compared to o-NeRF, our model (OSF) is able to disentangle lighting-dependent and view-dependent appearance and can render shadows.

These results suggest that OSF outperforms all baselines and ablations, both quantitatively and qualitatively. As in the previous experiment (Section 5.1), we find that OSF reproduces object appearances and self-shadows more accurately than the baselines. The difference is especially apparent in the couches in scenes (a) and (b), where the couches predicted by o-NeRF are extremely dark. This is due to the fact that o-NeRF is unable to disentangle view-dependence appearance from light-dependent appearance, and simply interpolates the radiance field learned another different lighting configuration. Please note that OSF is able to model inter-object light transport effects by rendering shadows cast by one object onto another and on the ground plane. Plus, it is able to render indirect illumination of one object reflecting light onto another. For example, light reflected from the left wall causes the left of the couch and table in scenes (a) and (b) to be brighter. Neither of these lighting effects are present in the o-NeRF results.

## 5.3 REAL-WORLD SCENES

In this section we evaluate our method on real world objects and scenes from the REAL-NRF and REAL-OUTDOOR datasets. For these experiments, we train one OSF for each object in REAL-NRF and each scene in REAL-OUTDOOR.

Figure 7 shows a comparison between ground truth, our method (OSF), and Neural Reflectance Fields (NRF) (Bi et al., 2020a). We show that OSF recovers stronger, more accurate specular highlights compared to NRF. OSF also produces more detailed appearances (see pony logo). This comparison demonstrates the main advantage of OSF: the ability to handle complex scattering functions.

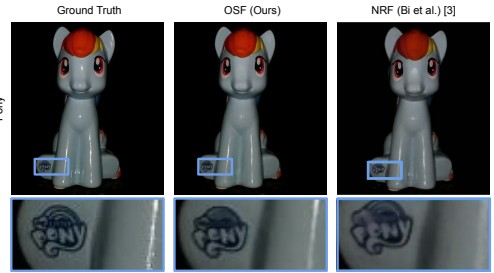

Figure 7: Comparison of OSF (ours) to Neural Reflectance Fields (NRF) (Bi et al., 2020a). OSF produces stronger, more accurate specular highlights on the legs (see zoomed view) and recovers more detailed appearances (see pony logo).

For scene composition, the OSFs trained on each object are composed with a synthetic floor OSF in Figure 8 row (a). Our method is able to compute accurate shadows, such as the shadow cast by the pony onto the two other objects in the scene. The indirect reflections from the floor allow the shadowed objects to be slightly visible as shown in the "OSF" panel.

Figure 8 rows (b) and (c) show results on inserting REAL-NRF objects into real outdoor scenes (REAL-OUTDOOR). Shadows and reflections are rendered with randomized lighting directions to approximate the environment lighting. Our method accurately renders occlusions between the inserted objects and the vase in Figure 8 row (c). Due to the compositional nature of OSFs, we are able to insert the learned pinecone from Figure 8 (b) into (c).

In Figure 8, each column shows ablated versions of OSF to study the impact of computing shadows and indirect illumination with our path tracing algorithm. "No Shadows, No Indirect" represents a version of our model containing only direct illumination (without modeling inter-object lighting effects). We additionally show "No Indirect" and "Indirect Only" variants of our model which

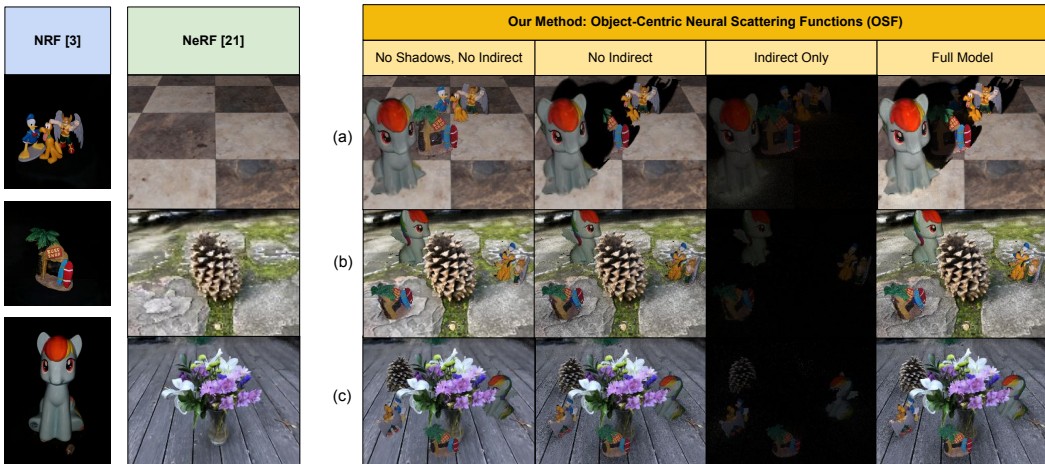

Figure 8: Real-world results. NRF (Bi et al., 2020a) and NeRF (Mildenhall et al., 2020) learn on individual static scenes or objects. In contrast, we compose real-world objects and scenes using OSFs. The objects are composed with a (a) synthetic floor and (b, c) real outdoor scenes from REAL-OUTDOOR. Columns show different ablated versions of our model: "No Shadows, No Indirect" which considers only direct illumination; "No Indirect" which includes both direct illumination and shadows; "Indirect Only" which considers only indirect illumination. Our OSFs show the most realistic renderings, with accurate shadows (e.g., pony shadowing the two other objects (row a) and indirect illumination (i.e., the ground and environment illuminating the objects).

represent computing shadows and indirect illumination, respectively. As illustrated by Figure 8, our full model containing both shadows and indirect illumination effects is the most realistic. Additional results on real-world scenes, including complex shadows, can be found in Appendix A.

## 6  DISCUSSION

We have proposed Object-Centric Neural Scattering Functions (OSFs), a method that enables composing objects captured only from photographs into photorealistic renderings of dynamic scenes. We demonstrated that decomposing a scene into implicit object functions that are view- and light-dependent enables reusabiliy of objects across scenes where objects, camera, and lighting can change. We presented a method for integrating our learned implicit functions with volumetric path tracing, and showed inter-object light transport effects such as shadow and indirect illumination for real-world objects where no computer graphics model is available. We believe our work is a step towards a graphics pipeline where real-world scenes are modeled by a composition of implicit functions to combine the flexibility of object-centric neural modeling with the photorealism of graphics rendering algorithms.

There are a few main limitations to OSF. First, the computational complexity of our method is high, but there are several works tackling the orthogonal issue of improving NeRF efficiency (as discussed in Section 4.2) that can easily be applied to OSFs. Second, while learning intra-object light transport means that intra-object path tracing is not needed, this formulation assumes that at test time, there are no occluders or light sources that intrude the object's convex hull (Sloan et al., 2002) (e.g., a person sitting in a chair). However, OSFs can still be rendered even if their bounding boxes are intersecting, as long as this assumption is not violated. Finally, acquiring datasets of real world objects with varying point light sources and viewpoints is challenging, but we hope that in the future such acquisition of real world datasets will become easier to capture and more widely available.

## REPRODUCIBILITY STATEMENT

We describe our method (Section 4) and experimental setup (Section 5) in detail to maximize reproducibility. We will release our code upon publication to facilitate future research.

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

## A REAL-WORLD SCENE COMPOSITION

Different scene configurations of composed objects from REAL-NRF are shown in Figure 9. We show the effect of moving the light, camera, or objects. Notice how the the appearance and shadows of the objects are updated across different scene configurations. Also notice that even when parts of the palm tree object and the cartoon object are cast under the pony's shadow, they do not appear completely dark due to the indirect illumination from the floor.

Analyzing the effect of different numbers of indirect (secondary) rays per primary sample, Figure 10 shows the result. As can be seen from the figure, the noisiness of the indirect illumination render decreases as the number of samples increase. Results in this paper contain between one and five randomly sampled secondary ray for each primary ray sample.

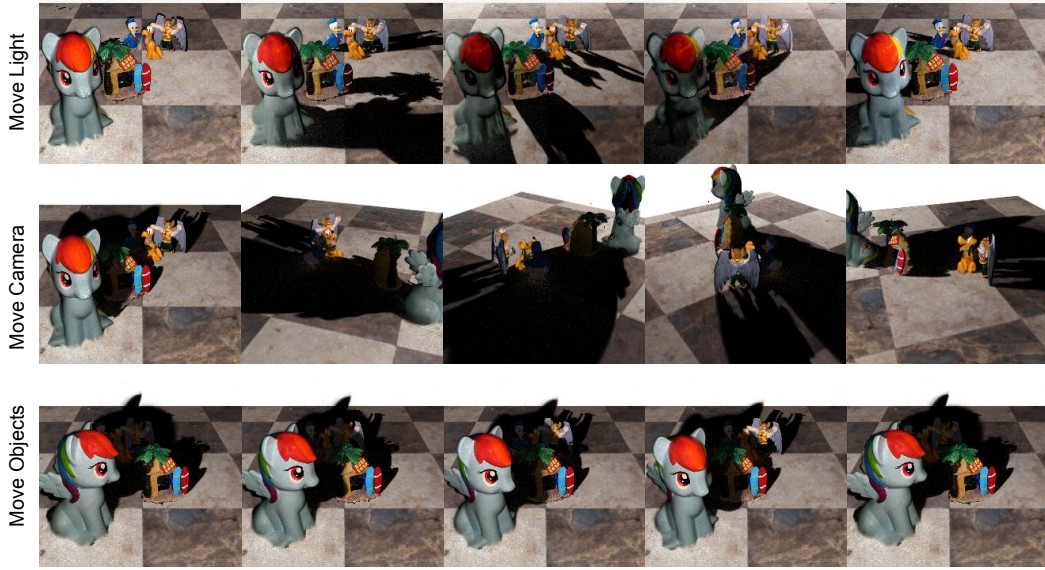

Figure 9: Composing real-world objects from REAL-NRF using our OSF method. We demonstrate the effect of moving the light, camera, or objects. Note how the appearance and shadows of the objects are updated across different scene configurations. Also notice that even when parts of the palm tree object and the cartoon object are cast under the pony's shadow, they do not appear completely dark due to the indirect illumination from the floor.

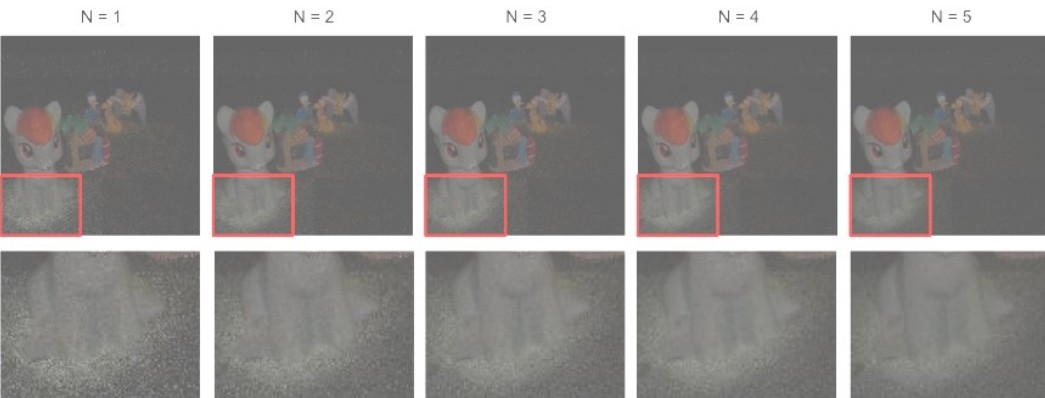

Figure 10: Visualizing the effect of different numbers of indirect (secondary) rays ($N$) per primary sample for our OSF model (the brightness of these images has been increased only for visualization purposes). Note that the noisiness of the render decreases as $N$ increases. We find that we are able to achieve relatively non-noisy results with approximately five samples.

Single-object renderings from REAL-NRF are shown in Figure 11. The objects were captured in a dark room with a one-light-at-a-time setup. After training OSF on each object in this dataset, we are able to render the objects from novel viewpoints and lighting directions.

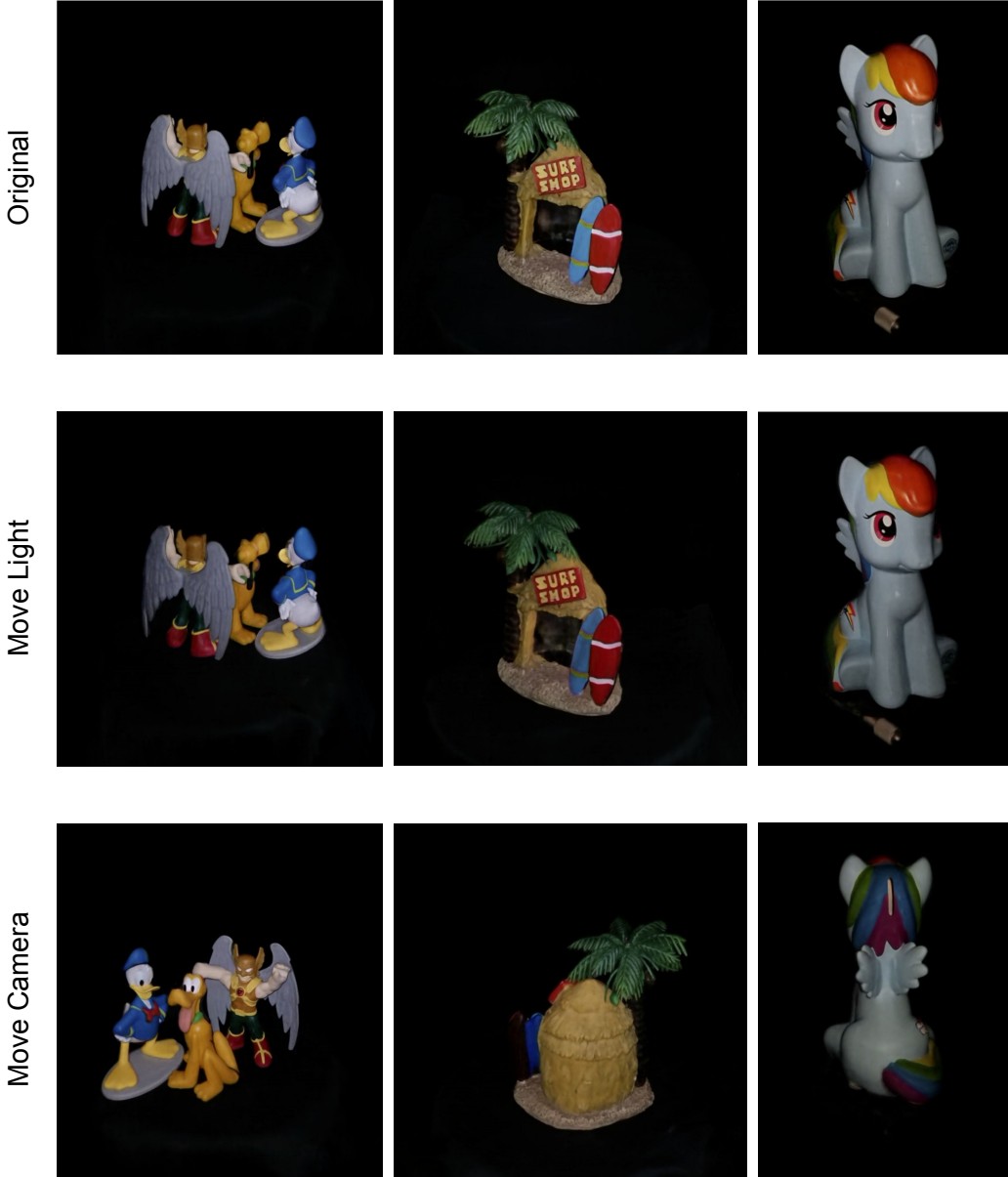

Figure 11: Learned OSFs on objects from REAL-NRF. The objects were captured in a dark room with a one-light-at-a-time setup. After training OSF on each object in this dataset, we are able to render the objects from novel viewpoints and lighting directions.

# B ABLATION EXPERIMENTS

Direct Only      Indirect Only      Direct + Shadows      OSF (Ours)

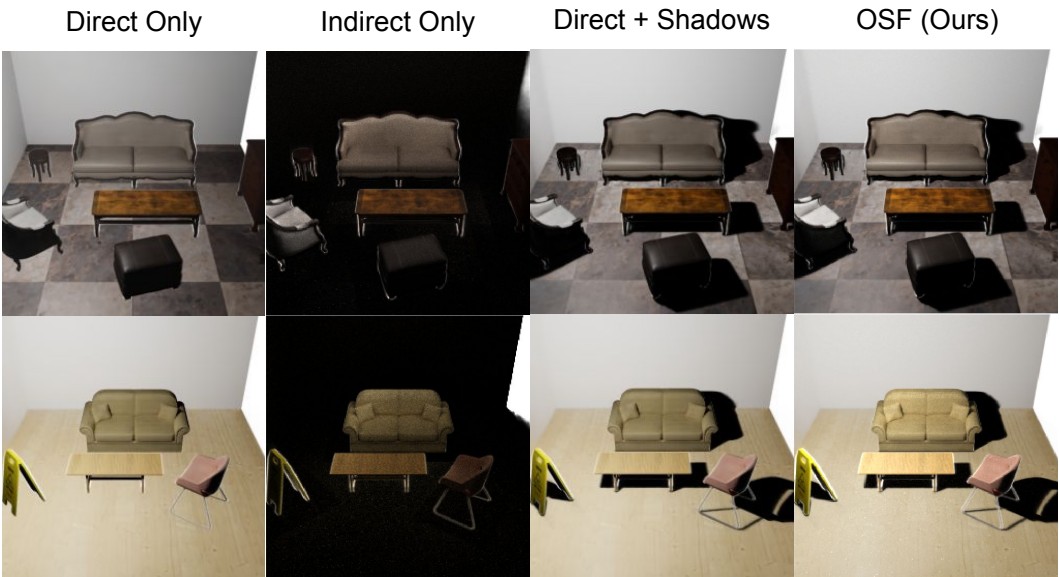

Figure 12: Ablation results on our OSF model.

Figure 12 shows ablation results on FURNITURE-REALISTIC. We evaluate different variants of our model: "Direct Only" which considers only direct illumination; "Indirect Only" which considers only indirect illumination; "Direct + Shadows" which includes both direct illumination and shadows. Our full model (OSF) shows the most realistic rendering, with accurate shadows and indirect illumination effects such as the left side of the couches and tables appearing brighter due to indirect lighting from the left wall. Note that the white area on the right of the images represent rays with zero density that are composited onto a white background (and therefore do not contribute indirect illumination to the scene).

OSF (ours)      o-NeRF + S      o-NeRF      Ground Truth

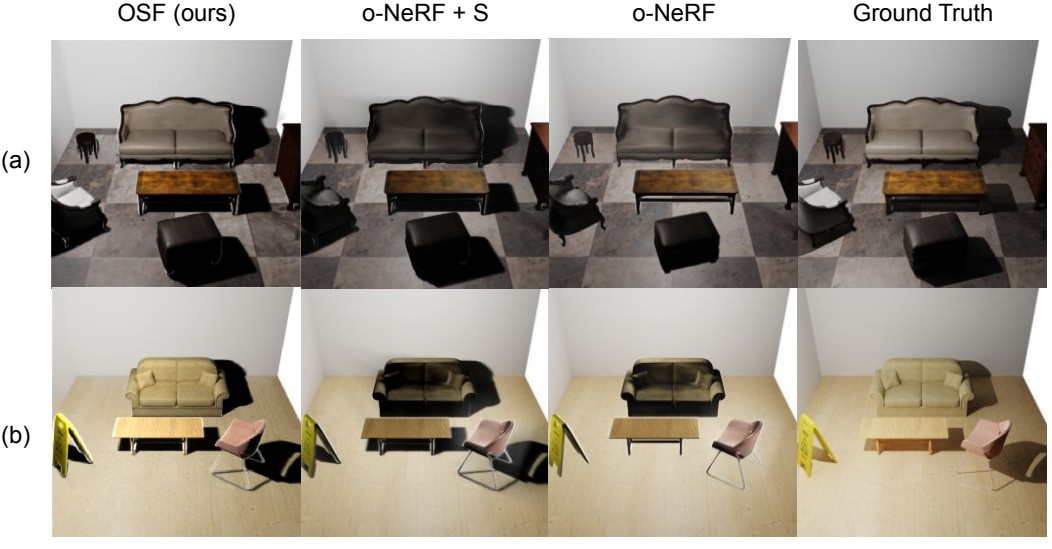

Figure 13: Comparisons on scene composition on FURNITURE-REALISTIC. The models OSF, o-NeRF, and o-NeRF + S are explained in §5. Compared to o-NeRF, our model (OSF) is able to disentangle lighting-dependent appearance from view-dependent appearance for individual objects, and is able to render shadows cast by objects onto the ground correctly.

## C   COMPLEX ILLUMINATION

In this experiment, we investigate how scenes composed of OSF objects can be rendered with complex illumination from an environment map.

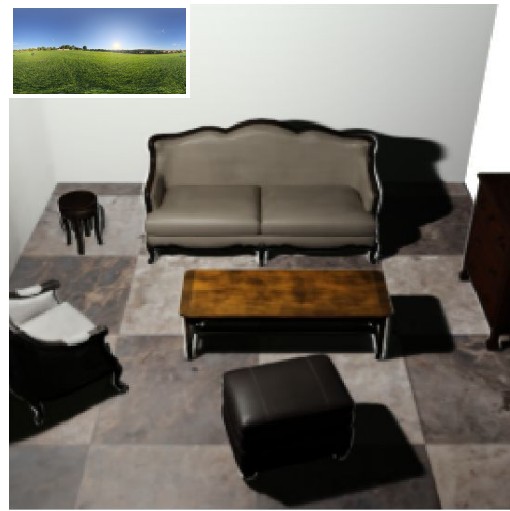

Specifically, we apply the combination of a point light source and the environment map shown in the top-left corner of Figure 14 to light one of our scenes in FURNITURE-REALISTIC. This simulates the appearance of the scene as if the scene were inserted into a complex lighting environment, which stresses the benefits of the OSF path tracing framework.

For each OSF sample point, we project the equirectangular coordinates of the environment map into spherical coordinates, sample 20 directions on the unit sphere uniformly at random, evaluate the OSF function for each incoming direction, and integrate them outgoing radiance using Equation 5. Please note that a green-blue tint is slightly apparent in the scene rendering, due to the contribution of green and blue lighting from the environment map.

Figure 14: Complex illumination results.

## D IMPLEMENTATION DETAILS

A flowchart of our method is shown in Figure 15.

We approximate our model $F_\Theta$ with a multilayer perception (MLP) with rectified linear activations. The predicted density $\sigma$ is view-invariant, while the scattering function value $\rho$ is dependent on the incoming and outgoing light directions. We use an eight-layer MLP with 256 channels to predict $\sigma$, and a four-layer MLP with 128 channels to predict $\rho$. For positional encoding, we use $W = 10$ to encode the position $x$ and $W = 4$ to encode the incoming and outgoing directions $(\omega_l, \omega_o)$, where $W$ is the highest frequency level. To avoid $\rho$ from saturating in training, we adopt a scaled sigmoid (Brock et al., 2016) defined as $S'(\rho) = \delta(S(\rho) - 0.5) + 0.5$ with $\delta = 1.2$. We use a batch size of 4,096 rays.

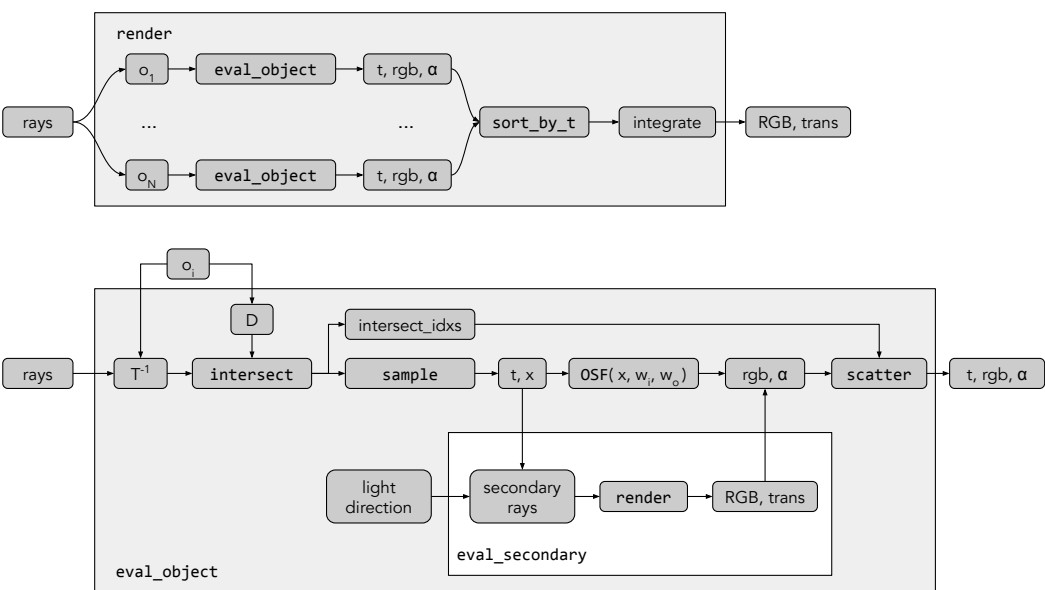

Figure 15: Flowchart of our method. See §4 for more details.

For synthetic datasets, we sample $N_c = 64$ coarse samples and $N_f = 128$ fine samples per ray. For real world datasets, we sample $N_c = 64$ coarse samples and $N_f = 64$ fine samples per ray. We use the Adam optimizer (Kingma & Ba, 2014) with a learning rate of 0.001, $\beta_1 = 0.9$, $\beta_2 = 0.999$, and $\epsilon = 10^{-7}$.

