# OpenReview forum: "Object-Centric Neural Scene Rendering"
_ICLR.cc/2022/Conference — ICLR 2022 Submitted_

### Official Review · Reviewer_6eq6 · 2021-10-23

**Correctness:** 4
**Technical Novelty And Significance:** 3
**Empirical Novelty And Significance:** 1
**Recommendation:** 5
**Confidence:** 4

**Main Review:**

Pros:

The method is well-designed and models even indirect illumination. This is a significant and important contribution. It enables a decent level of control over neural scene representations, namely rigid transforms of objects, while properly/explicitly modelling the resulting appearance changes via path tracing. While extending neural radiance fields to object-level rigid transformations is rather straightforward, accounting for the illumination is not.

There are some results on real scenes.

------

Cons:

Colored illumination:
- Direct: Shadows rays (Equation 4) assume that all color channels are absorbed/occluded equally? So colored glass for example would cast a grey shadow and not a colored shadow? Were there any experiments to account for such effect, e.g. per-color-channel densities?
- Indirect: Indirect illumination should be colored if I understand the method correctly? I cannot see any result though that exhibits colored indirect illumination, not even for the yellow sign in Fig. 12.
- In both cases, a result showing that colored illumination is possible or an explicit statement that color is not taken into account is necessary I believe (e.g. in limitations or as a stated assumption when introducing the method).

Runtime:
- An evaluation of the runtime (training time and inference/rendering time) is missing. It could be measured in dependence on the number of objects (1,2,3,5 for example) and number of bounces (1,2,3 for example), M could be kept fixed, rays could be sampled randomly instead of rendering full images if full images are too expensive. I believe that a comparison to NeRF in that regard is also necessary to give an understanding of how much impact ray tracing has on runtime (number of objects=1 and bounces=1 should cover this case, I believe?). Such an experiment would also provide experimental support for the claim at the end of page 6 regarding runtime.
- Related to that, an experiment (qualitative and quantitative) where the number of randomized light directions for indirect lighting (and also environment map sampling; number of directions: few, normal, many), number of points on each secondary ray (M; few, normal, many) and the recursion depth/number of bounces (few, normal, many) varies would be ideal. This could also be done with smaller image patches (say 50x50 pixels) if full images are too slow. That would give an idea on what the trade-offs are between fidelity, runtime, and noise. If compute is still too high, each of the three axes could be explored independently (with the other two axes set to normal) and then also jointly (all three axes set to few, all three set to normal, all three set to many).

Consistency:
A supplemental video would have been useful, e.g. for switching between different lighting conditions. It could also have shown whether the scene transforms smoothly/nicely/consistently when moving the light around or when moving the camera. Still images would be insufficient to demonstrate this.

Limitations:
Limitations are missing. Could mention again that objects' bounding boxes are not allowed to intersect. Runtime and only grey-scale/uncolored direct shadows are also likely limitations.

Experiments:
Experiments are only on a few scenes.

Reproducibility:
There is no mention of a code release.

------

Not relevant for the decision:

Minor questions:

- Fig. 12: Why are the floor and walls black for the indirect-only setting? Are they not just treated like objects? Also, is the visualization of the indirect-only result exaggerated or why is the side of the white chair visibly white in indirect-only but not in OSF?
- The training settings are not sufficiently specified, the new parameters are missing: number of bounces and number of points sampled on secondary and shadow rays. The number of training iterations per object is also missing. The resulting training time both on synthetic and real-world objects is also not mentioned.
- first line page 3: "especially for objects with intersecting bounding volumes." -- This method shares this weakness, right?
- Sec. 4.2: just for clarity, is each object's bounding box (for ray intersection) computation axis-aligned with the object coordinate system or the world/scene coordinate system?
- Is there anything that constrains (in a soft or hard manner) the outgoing fractions to sum up/integrate to 1 or at most 1 for a given incoming light direction?
- Fig. 10: What exactly is N in this figure? N is used in the main text to refer to the number of objects and to the number of point samples along a ray, neither of which seems like the right parameter here.

Minor suggestions for improvements:

- Fig. 7: I currently cannot see much in this figure, a comparison to a white/grey environment map would make it easier to tell that there is an effect.
- I'm not a fan of the equation two lines after Eq. 4. I understand what it's trying to say but I believe this needs to be changed to be mathematically correct, unless that makes a bunch of other equations messy. Also, why is it L_l instead of just L? That notation should be introduced beforehand.
- Fig. 8: Switching out columns 2 and 3 would make the difficult comparison between No Indirect and Full Model easier.
- There's a typo at the end of page 2: from from

**Summary Of The Paper:**

The paper proposes a decomposition of a 3D scene into object-level neural radiance fields. This allows to apply rigid transforms to each object independently and to rearrange the scene. Crucially, light transport is modelled, which turns the radiance fields into scattering functions. Direct lighting and shadowing, as well as indirect illumination from several light bounces are taken into account, such that the illumination of the modified scene looks correct. As long as the scattering fields are trained on a sufficient number of light positions, the lights can also be moved around at test time.

**Summary Of The Review:**

The paper proposes a well-designed method and is promising. However, experiments are not yet quite sufficient for acceptance. That's why I will go with borderline reject for now.

Experiments on more real scenes, for example from the same dataset as currently, should be added.

Furthermore, the experiments suggested under "Runtime" above should be added, though possibly in a reduced form since the method might be too slow to feasibly obtain these numbers. In that case, I'd prefer an experimental setup wherer the number of pixels is reduced (e.g. randomly picking 1000 rays from a full image, and doing that for, say, 100 images, ideally across different scenes in terms of objects and object positions) rather than fewer settings tried.

Demonstrating plausible consistency while smoothly changing light position/object positions also strikes me as beneficial but I don't believe that acceptance should hinge on it.

---

> ### Author Response · Authors · 2021-11-23
> **Response to Reviewer 1**
>
> Thank you for your constructive review and helpful suggestions!
>
> > Shadows rays (Equation 4) assume that all color channels are absorbed/occluded equally? So colored glass for example would cast a grey shadow and not a colored shadow? Were there any experiments to account for such effect, e.g. per-color-channel densities?
>
> No. Colored glass would cast a colored shadow. Equation 4 simply computes the transmittance along a shadow ray, independent of the color of the light traveling along the ray. This is later multiplied by the “color” of the ray in Equation 5. So the color of light traveling through colored glass is unchanged, but the lower transmittance properties of glass attenuate the amount of light that travels through.
>
> > ...an experiment (qualitative and quantitative) where the number of randomized light directions for indirect lighting (and also environment map sampling; number of directions: few, normal, many), number of points on each secondary ray (M; few, normal, many) and the recursion depth/number of bounces (few, normal, many) varies would be ideal
>
> Thanks for the suggestion. We have included a qualitative analysis of the effect of the number of secondary rays compared to the noise level in the rendered image in Figure 11 of the Supplementary Material.
>
> > A supplemental video would have been useful, e.g. for switching between different lighting conditions. It could also have shown whether the scene transforms smoothly/nicely/consistently when moving the light around or when moving the camera. Still images would be insufficient to demonstrate this.
>
> These are great suggestions. We have attached a supplementary video showing animations of changing light, viewpoint, and object poses.
>
> > There is no mention of a code release.
>
> We have added a reproducibility statement in the paper.
>
> > Limitations: Could mention again that objects' bounding boxes are not allowed to intersect. Runtime and only grey-scale/uncolored direct shadows are also likely limitations.
>
> We have added a discussion of the limitations in Section 6 of the paper. However, we’d like to point out that object bounding boxes *are* in fact allowed to intersect. Even if you have intersecting bounding boxes, as long as there are no occluders or light sources within the object’s convex hull (which would change how light bounces in that object), OSF still works. Colored shadows are also supported, as we explained in our earlier response.
>
> > Evaluation of the runtime (training time and inference/rendering time)
>
> Thanks. We have added a discussion on runtime at the end of Section 4.2.
>
> We hope that our responses are helpful in addressing your concerns. Please don't hesitate to let us know if there are any additional questions.

---

> > ### Comment · Reviewer_6eq6 · 2021-11-29
> > **Keeping my rating**
> >
> > I will keep my original rating. I cannot judge the trade-off between runtime and quality/noise, and the number of evaluated scenes, given that they are mostly synthetic, is too limited. I appreciate the additional details on some of the other aspects, especially the supplemental video.

---

### Official Review · Reviewer_TPuY · 2021-10-26

**Correctness:** 3
**Technical Novelty And Significance:** 3
**Empirical Novelty And Significance:** 2
**Recommendation:** 5
**Confidence:** 3

**Main Review:**

Strengths:
+ This paper introduces 7D object-centric neural scattering functions (OSFs) using a lighting- and view-dependent network. The additional light direction input makes this representation support novel lighting rendering.
+ Given multiple optimized OSFs, this paper proposes to render a compositional scene using volumetric path tracing, light transport effects like occlusions, specularities, shadows, and indirect illumination are considered.
+ Some of the visual results in the paper look visually pleasing.

Weaknesses:
- The proposed OSFs sound interesting. However, it requires hundreds of images captured under different point light sources and viewpoints for optimizing the network, which is difficult to achieve in the real world.
- The proposed OSFs representation is not well validated. The baseline (o-NeRF) compared in the paper is not very convincing. Comparison with existing appearance modeling methods (e.g., Neural Reflectance Fields, NeRV) on single object rendering under novel views and lightings should be provided.
- Volumetric path tracing for multiple objects rendering is not new.
- The rendering time complexity seems to be very high. However, no discussion is provided in the paper.


**Summary Of The Paper:**

This paper proposes a NeRF based method for composing photo-realistic scenes from captured images. The proposed method learns object-centric neural scattering functions (OSFs) to implicitly model per-object light transport using a lighting- and view-dependent neural network. Multiple objects can be rendered with volumetric path tracing. The proposed method has been evaluated on both synthetic and real-world datasets.

**Summary Of The Review:**

This paper introduces object-centric neural scattering functions for modeling light transport of an object, allowing novel lighting rendering and scene composition. However, the data requires to optimize the OSFs is restricted and the rendering time complexity is high. Also, the experiment is not very convincing due to the lack of comparisons. I would like to give a negative rating at the current stage.

---

> ### Author Response · Authors · 2021-11-23
> **Response to Reviewer 2**
>
> Thank you for your constructive review and helpful suggestions!
>
> > Comparison with existing appearance modeling methods
>
> We agree! We have updated the paper with a comparison to Neural Reflectance Fields (Bi et al. 2020a) in Section 5.4 and Figure 8.
>
> > Rendering time complexity
>
> Thanks for the suggestion. We have added a discussion on runtime at the end of Section 4.2.
>
> We hope that our responses are helpful in addressing your concerns. Please don't hesitate to let us know if there are any additional questions.

---

> > ### Comment · Reviewer_TPuY · 2021-11-29
> > **Feedback for the authors**
> >
> > Thanks for the author's responses. After checking the comments of other reviewers and the authors' responses, I feel that the limited novelty, lack of enough comparison, and the runtime inefficiency are common concerns. I would like to keep my original rating of "marginally below the acceptance threshold".

---

### Official Review · Reviewer_KXhU · 2021-11-01

**Correctness:** 3
**Technical Novelty And Significance:** 2
**Empirical Novelty And Significance:** 2
**Recommendation:** 5
**Confidence:** 3

**Main Review:**

Strength:
1. The idea of representing object as an light transportation function is a novel contribution, prior works are operating on predifined BRDF function with limited capability.
2. The paper also proposed an approach to render with implicit function with secondary lighting effect and shadows while compositing the scene.
3. The paper is well written and easy to follow.

Weakness:

1. In the paper, to train the OSFs, the authors assumes the object is captured with the point light with radiance of (1, 1, 1), is the position of this point light is known as well? or the pipeline can also backwards to estimate the position of point light? This assumption (even just the point light with fixed radiance and unknown position)  is hard for real object, as the real illumination can be arbitrary complex.

2. For OSFs, The author argues "our method is capable of learning all scattering functions, and can render multiple objects in dynamic scenes." I agree with this argument, as using a MLP to represent light scattering function have a higher capacity than pre-defined BRDF formulation. However, it is missing in the experiment to demonstrate this, as there is no comparison on why using MLP to parameterize lighting scattering function is better than others (e.g. Nerual reflectance field). There are also other works to estimate the light scattering function: Nerd [a], PhySG [b], which uses spherical Gaussian to represent the light, and Disney BRDF. I suggest the author to compare with Neural Reflectance Fields (Bi et al., 2020a) or PhySG on experiment Sec. 5.1 to demonstrate this point.

3. Computation cost of the whole process is huge. In particular, for each pixel, the paper sample M points, and for each point, the paper sample K directions in a sphere and for each direction, the the author further sample M points to estimate the intensity for each object, this gives M*M*K*(N-1)*(N_pixel) in total, could the author provide the computation cost comparison for it? (e.g. memory and rendering time)

4. minor points: in Fig. 7, it's very hard to see the green-blue tint as mentioned by the author, it would be better if the author can show a comparisons with point light to see the effect of rendering with environment map.

5.  The proposed the rendering technique for secondary lighting effect (shadow) is great, however, in principle, the proposed approach should also be able to backwards through this? it would be better if the paper can show some results of using it to run backward optimization.

[a] NeRD: Neural Reflectance Decomposition from Image Collections
Mark Boss, Raphael Braun, Varun Jampani, Jonathan T. Barron, Ce Liu, Hendrik P. A. Lensch

[b]PhySG: Inverse Rendering with Spherical Gaussians for Physics-based Material Editing and Relighting
CVPR 2021
Kai Zhang*  Fujun Luan*  Qianqian Wang  Kavita Bala  Noah Snavely

**Summary Of The Paper:**

The paper proposed a method to composite objects parameterized using implicit functions into realistic scenes. The idea is to first capture the representation for each object separately, and each object is represented using neural scattering function, which predicts the outgoing lighting transport conditioned on the input lighting direction, viewing direction and 3d location. The objects are composed into the full scene by doing volume rendering along the ray, and the radiance of each sampled point in the ray is calculated via integrating on a sphere to obtain the lighting radiance (including secondary (indirect) light effect, the shadow effect is obtained from iterating the ray from light position to the  sampled point). The results on both synthetic scene and real scene shows improvement over baselines (o-Nerf, o-Nerf S) that didn't consider the lighting transportation.

**Summary Of The Review:**

In summary, the proposed method using lighting transportation function is a novel approach to represent an object and extends previous works on learning BRDF of an object, considering the missing comparisons and the limitations, I vote for the board line initially, and I'm happy to listen to the authors and other reviewers.

---

> ### Author Response · Authors · 2021-11-23
> **Response to Reviewer 3**
>
> Thank you for your constructive review and helpful suggestions!
>
> > Comparison on why using MLP to parameterize lighting scattering function is better than others (e.g. Nerual reflectance field).
>
> Thanks for the suggestion. We have updated the paper with a comparison to Neural Reflectance Fields (Bi et al. 2020a) in Section 5.4 and Figure 8.
>
> > Computational cost
>
> We have added a discussion on runtime at the end of Section 4.2.
>
> > Assumption on knowing the position of the point light
>
> We assume that the position of the light source is known. While it is difficult to capture real world objects in arbitrary scenes, we can capture real world objects with a light stage-like setup, similar to the real world objects evaluated in the paper.
>
> We hope that our responses are helpful in addressing your concerns. Please don't hesitate to let us know if there are any additional questions.

---

### Official Review · Reviewer_4NH2 · 2021-11-02

**Correctness:** 2
**Technical Novelty And Significance:** 2
**Empirical Novelty And Significance:** 2
**Recommendation:** 5
**Confidence:** 4

**Details Of Ethics Concerns:**

I have no ethics concerns with regard to this paper.

**Main Review:**

In my opinion, the strengths of the paper include:

1. It can be beneficial to represent objects as neural scattering functions rather than neural reflectance fields. As pointed out in the paper, this may allow the modeling of more complex materials. Besides, it may have the potential to support fast rendering of in-object indirect illumination. However, I feel both benefits are not clearly demonstrated in the paper.
2. The proposed method demonstrates clear improvement with respect to classical baseline Nerf.
3. The rendering pipeline considers relatively complex light transport such as soft shadows and indirect reflection, which can be considered as an improvement compared to many prior works.

There are probably some questions needed to be classified, which are listed below:
1. The benefit of the representation may not be fully demonstrated.
The method that is most closely related to the proposed method should be neural reflectance fields by Bi et al. The major benefit of the proposed method may include the potential to handle more complex materials and the ability to fast render in object indirect illumination. However, both benefits are not clearly demonstrated in the paper, which makes the arguments weak.

2. The rendering equation may be biased.
I am a little concerned about how shadows are rendered in this paper. According to equation 4, it simply computes the product of the alpha values of all samples. However, here the samples are not uniformly created for every ray or every object because authors assign the same number of samples for every object, which means smaller object will have higher sampling rate . This may lead to bias when rendering shadows, especially soft shadows. I feel this is an issue that should probably be corrected.

3. The novelty of the paper may be limited.
The three contributions mentioned in the introduction can probably be summarized into 2 major contributions. One is the novel representation. However, there is no comparison with the prior representation (neural reflectance fields) to demonstrate that the new representation is better. Therefore this contribution may not be fully supported. The other is the rendering algorithm. However, the proposed algorithm seems to be a simplified version of standard volume path tracing, without importance sampling of the density field and the light sources.  Besides, prior work (neural reflectance fields) has already demonstrated that their representation can be rendered by a standard renderer using volume path tracing (See Fig 9 in neural reflectance fields). Therefore, the second contribution may not be considered as very novel and significant either.

4. Necessary comparisons may be missing.
Besides neural reflectance fields, authors may consider comparisons with other neural implicit representations that support relighting, including PhySG by Zhang et al. NerD by Boss et al., and NeRFactor by Zhang et al. In addition, it will be better to compare with traditional representation using mesh and BRDF. The state-of-the-art can be Luan et al. EGSR 2021.

Minor issues:
1. Notations can be cleaned.
Several notations may never be used. That may include $\gamma$ in the second paragraph of section 4.2 and $\Gamma$ in the third paragraph of section 4.2. Meanwhile, $\mathbf{r}(t)$ in the second paragraph of section 4.2 comes out without being explained first.

2. Citations and related works can probably be improved. Since this paper focuses on scene compositing, it may be better to have one more paragraph in related work discussing recent scene compositing methods. That includes neural implicit methods like GIRAFFE by Niemeyer et al. and more traditional methods like Lighthouse by Srinivasan et al. and Inverse rendering by Li et al.









**Summary Of The Paper:**

This paper proposes a new neural implicit representation called object-centric scattering function for scene compositing application. The major extension is to add the lighting direction as an input so that the new representation can be used for relighting. To train this representation, the authors propose minimizing the rendering loss of images rendered from different views under different point lighting. Given this representation, the authors propose a standard volume path tracing framework to render different objects and scene structures together, with indirect illumination and shadow being modeled. Experiments show that compared with nerf, the proposed method achieves better accuracies in object compositing and can handle changes of illumination.

**Summary Of The Review:**

Given the above issues, my current review leans towards negative. My major concern is the lacking of novelties and the technique correctness of some design choices. I will appreciate it if authors can address my concerns in the rebuttal and I will change my reviews accordingly.

---

> ### Author Response · Authors · 2021-11-23
> **Response to Reviewer 4**
>
> Thank you for your constructive review and helpful suggestions!
>
> > Comparison with the prior representation (neural reflectance fields).
>
> We have updated the paper with a comparison to Neural Reflectance Fields (Bi et al. 2020a) in Section 5.4 and Figure 8. As you correctly pointed out, the results show that OSF is capable of handling more complex materials compared to neural reflectance fields.
>
> > Relation to standard volume path tracing and prior work on neural reflectance fields
>
> This is a fair point. But up until now, only parametric models (e.g., BRDF) have been explored, as in neural reflectance fields. This assumes knowledge of the underlying parameterization to be optimized. In contrast, our work introduces a more generic formulation that can handle any scattering function (including complex, real-world scattering phenomena) that is unable to be captured with BRDF models. Furthermore, using a BRDF representation requires running volumetric path tracing *within* each object to simulate intra-object light bounces. Our formulation does not require this and is therefore significantly faster to render.
>
> > Samples are not uniformly created for every ray or every object.
>
> Correct, smaller objects have a higher “sampling rate” compared to large objects. This can easily be normalized by dynamically computing the number of samples based on the length of the line segment from ray-bounding box intersections.
>
> We hope that our responses are helpful in addressing your concerns. Please don't hesitate to let us know if there are any additional questions.

---

> > ### Comment · Reviewer_4NH2 · 2021-11-29
> > **Change the rating to marginally below the threshold**
> >
> > Thanks authors for adding the new comparisons and the rebuttal! It partially solves my questions on comparisons with prior works and also comparisons to standard volume path tracing. However, given the results in Figure 8, I feel it is hard to say that the new representation can achieve higher quality compared to NRF. Some highlights are missing in the proposed method. Therefore, I feel authors should probably consider arguing more on the efficiency of the new method (No intra object path tracing), rather than on the rendering quality. I will change my rating to marginally below the threshold.

---

### Decision · Program_Chairs · 2022-01-20

**Decision:**

Reject

**Comment:**

All three reviewers recommend borderline rejection based on limited novelty, missing comparisons with other methods, and runtime inefficiency. The authors’ response helped clarify other questions but did not eliminate the main concerns about the paper. The AC agrees with the reviewers that, in its current form, the paper does not pass the acceptance bar of ICLR. The reviews have detailed comments and suggestions that should help the authors to improve the work for another conference.